# Material Design for Optimal Postbuckling Behaviour of Composite Shells

**DOI:** 10.3390/ma14071665

**Published:** 2021-03-28

**Authors:** Domenico Magisano, Francesco Liguori, Antonio Madeo, Leonardo Leonetti, Giovanni Garcea

**Affiliations:** 1Dipartimento di Ingegneria Informatica, Modellistica, Elettronica e Sistemistica, University of Calabria, 87030 Rende, Italy; domenico.magisano@unical.it (D.M.); francesco.liguori@unical.it (F.L.); antonio.madeo81@unical.it (A.M.); leonardo.leonetti@unical.it (L.L.); 2CIRTech Institute, Ho Chi Minh City University of Technology (HUTECH), Ho Chi Minh City 725600, Vietnam

**Keywords:** composites, shells, post-buckling, optimisation, material design, Koiter method, imperfection sensitivity

## Abstract

Lightweight thin-walled structures are crucial for many engineering applications. Advanced manufacturing methods are enabling the realization of composite materials with spatially varying material properties. Variable angle tow fibre composites are a representative example, but also nanocomposites are opening new interesting possibilities. Taking advantage of these tunable materials requires the development of computational design methods. The failure of such structures is often dominated by buckling and can be very sensitive to material configuration and geometrical imperfections. This work is a review of the recent computational developments concerning the optimisation of the response of composite thin-walled structures prone to buckling, showing how baseline products with unstable behaviour can be transformed in stable ones operating safely in the post-buckling range. Four main aspects are discussed: mechanical and discrete models for composite shells, material parametrization and objective function definition, solution methods for tracing the load-displacement path and assessing the imperfection sensitivity, structural optimisation algorithms. A numerical example of optimal material design for a curved panel is also illustrated.

## 1. Introduction

Thin-walled composite structures are employed in a wide range of structural applications, particularly in the aerospace industry due to the high strength-to-weight ratio. Their design is dominated by buckling, which is mainly influenced by geometry and material properties. The geometry is usually constrained by the structural functionality and only little changes are possible. Conversely, the spatial distribution of the elastic properties (like for the fibre orientation) can be easily varied. Then, an optimization process of the material distribution can provide the desired mechanical behavior in terms of displacement and capacity. Different manufacturing options are also available to tailor the stiffness and reduce buckling effects: grid stiffeners [1], multi-layered and variable thickness composites [2], variable angle tows (VATs) [3]. New technologies for 3D printed products are also opening new prospective for cheaper and more tunable structures [4,5,6].

Moreover, as known, optimal properties in composites are usually sought also by controlling the orientation of the fibers in each layer, since the fibers orientation significantly affects the stiffness distribution, hence, the load-carrying capability or the elastic limits states (see, for instance, [7,8]). A promising direction in the context of material optimization is that offered by nanostructured materials which can exhibit multifunctional properties and are thus prone to more advanced multi-objective optimizations. In this field, nanocomposite materials made of thermosetting or thermoplastic polymers integrated with carbon nanotubes (CNTs) are currently subject to intense developments [9,10,11].

The mechanical behavior of shells can be simulated by different structural models. The Mindlin–Reissner model is the most common for shear flexible shells, where the kinematics is governed by displacements and rotations of the middle surface. The Kirchhoff–Love one is attractive for thin shells, where only displacements of the mid-surface are needed. Alternative shear flexible models have been proposed, like solid-shell elements, which are solid elements able to obtain the shell solution without meshing through the thickness. Such formulations use displacement degrees of freedom (DOFs) only and the number of overall DOFs can be equal to the one in Mindlin–Reissner elements [12,13], but without the rotation parametrization in large deformation problems. Beside finite element formulations [12], Isogeometric analysis (IGA) based on NURBS highly continuous shape functions [14] is an interesting alternative for the description of geometry and kinematics over the shell mid-surface.

The linearized buckling load is the objective function in many optimal design strategies proposed in the literature. However, this can lead to an elastic limit state known as buckling mode interaction, characterized by unstable post-critical behavior [15] with a deterioration of their capacity due to geometrical, load and material imperfections. Instead, a more reliable design, which takes into account the geometrically nonlinear behaviour, should be considered. In this case, the failure load of the structure can be used as objective function to maximize. This can be defined as the first limit load for the unstable load–displacement curves or as the load leading to deformation limits, taking into account the typical post-buckling stiffness reduction. Optimizing the post-buckling behaviour is a challenging task. Firstly, a suitable mechanical model and its discrete approximation are required to describe with acceptable accuracy geometry, boundary conditions and deformations. Discretization techniques are usually needed and, then, the structural problem is generally described by a high number of discrete nonlinear equations defining the equilibrium load-displacement path. Moreover, an imperfection sensitivity is generally needed for reliable estimates of the safety factor.

The static Riks method [16,17] is a standard tool for path following the solutions of a set of nonlinear equations. This approach is suitable for assigned data, but not for structural optimizations, which require a new equilibrium path for any change in the design variables, since the single run is too time-consuming with current CPUs. The same holds for an imperfection sensitivity analysis. Promising generalizations of the Riks method have been presented in [18,19,20], which are able to perform parametric analysis in a more efficient way by using the fold line concept.

In the optimal design presented in [21], the failure load is given by a nonlinear finite element (FE) buckling problem, extended in [22,23] in order to consider the worst case of the geometrical imperfection. An alternative reduced order model formulation is offered by numerical implementations of Koiter’s theory of elastic stability [24], allowing to estimate the initial post-critical behaviour in terms of slope and curvature of the bifurcated branches [25].

More recently, a solution algorithm based on Koiter’s theory implemented within a Finite Element environment was proposed in [26,27]. In this case, the design is able to consider general geometries, loading and boundary conditions. Moreover, a good accuracy in predicting the initial postbuckling response is given by a multi-modal asymptotic expansion which accounts also for nonlinear buckling modal interactions [28]. The strategy also provides an inexpensive sensitivity analysis with a statistical estimation of the worst-case imperfection, assumed to be a combination of the linearized buckling modes of the perfect structure. A hybrid solution strategy, referred to as the Koiter–Newton approach, was further investigated in [29,30].

Despite the difficulties associated with the prediction of the nonlinear behavior, another challenge is the solution of the optimization problem that is generally nonlinear and nonconvex. Its solution is usually computationally expensive and difficult due to the possibility of local minima. Frequently employed algorithms are the random search methods [26,31], genetic algorithms [32] and gradient-based techniques such as the method of moving asymptotes [33] or sequential linear programming [34].

This paper is a review of the recent computational developments concerning the material design for optimising the post-buckling response of composite thin-walled structures, showing how baseline products with unstable behaviour can be transformed in stable ones operating safely in the post-buckling range. Four main aspects are discussed: mechanical and discrete models for composite shells, material parametrization and objective function definition, solution methods for tracing the load–displacement path and assessing the imperfection sensitivity, structural optimisation algorithms. A numerical example of optimal material design is also given, where the optimisation suppresses the snap-through instability leading a globally stable behaviour. Examples of applicability for actual engineering uses can be found in [27,35].

The paper is organized as follows: Section 2 describes the solid-shell continuum and its discrete finite element and isogeometric counterpart for elastic shell structures; Section 3 reviews the material parametrizations and introduces the optimization problem; Section 4 discusses the solution method for tracing the equilibrium path of a slender elastic structure with particular focus on the reduced order modelling provided by the Koiter method; Section 5 reviews the nonlinear optimization algorithms suitable for the problem under consideration; a numerical example of optimization for a curved panel is illustrated in Section 6; finally, conclusions are drawn in Section 7.

## 2. Mechanical and Numerical Models for Composite Shells

The main equations of the shell model are now described. The outset is the 3D Cauchy continuum whose deformation is described by means of the Green–Lagrange strain. A linear approximation is assumed through the shell thickness for the kinematics according to the solid-shell formulation [12,25,36]. The model is rotation-free, so making simple and effective its discrete formulation. Two kinds of discretization over the shell mid-surface are presented, namely linear finite elements and isogeometric (IGA) formulations.

### 2.1. Kinematics of the Shell from the 3D Continuum

Convective curvilinear shell coordinates $\zeta =(\zeta_{1},\zeta_{2},\zeta_{3})$ are employed, with $\left(\zeta_{1},\;\zeta_{2}\right)$ representing mid-surface coordinates and $\zeta_{3}\in \left[-\frac{h}{2},\frac{h}{2}\right]$ being the thickness coordinate with *h* the shell thickness (see Figure 1). The current position vector p(ζ) is expressed in terms of the reference position vector x(ζ) and the displacement field u(ζ),
(1)p(ζ)=x(ζ)+u(ζ).

The covariant basis vectors in the undeformed configuration are $G_{i}=x{,}_{i}$, where $\left(\right){,}_{i}$ denotes partial differentiation with respect to the *i*th component of ζ. The contravariant basis $G^{i}$, so that $G_{i}\cdot G^{j}=\delta_{i}^{j}$ with $\delta_{i}^{j}$ the Kronecker delta and (·) the dot product, makes it possible to write the Green–Lagrange strain as
(2)$$E={\bar{E}}_{ij}\;G^{i}⊗G^{j},\;{\bar{E}}_{ij}=\frac{1}{2}\left(x{,}_{i}\cdot u{,}_{j}+u{,}_{i}\cdot x{,}_{j}+u{,}_{i}\cdot u{,}_{j}\right),$$
where (⊗) indicates the tensor product.

Assuming a linear interpolation along the thickness direction, the position vector becomes
(3)$$x\left(\zeta \right)=x_{0}\left(\zeta_{1},\zeta_{2}\right)+\frac{2\zeta_{3}}{h}x_{n}\left(\zeta_{1},\zeta_{2}\right)$$
where $x_{0}:=\frac{1}{2}\left(x\left(\zeta_{+}\right)+x\left(\zeta_{-}\right)\right)$ and $x_{n}:=\frac{1}{2}\left(x\left(\zeta_{+}\right)-x\left(\zeta_{-}\right)\right)$, with $\zeta_{+}=\left(\zeta_{1},\zeta_{2},\frac{h}{2}\right)$ and $\zeta_{-}=\left(\zeta_{1},\zeta_{2},-\frac{h}{2}\right)$. Similarly, the displacement field is described as
(4)$$u=u_{0}\left(\zeta_{1},\zeta_{2}\right)+\frac{2\zeta_{3}}{h}u_{n}\left(\zeta_{1},\zeta_{2}\right)$$
with $u_{0}:=\frac{1}{2}\left(u\left(\zeta_{+}\right)+u\left(\zeta_{-}\right)\right)$ and $u_{n}:=\frac{1}{2}\left(u\left(\zeta_{+}\right)-u\left(\zeta_{-}\right)\right)$ being the coordinates of the upper and lower surfaces of the shell. The independent strain components in Equation (Equation 2) are collected in the 6-dim strain vector $\epsilon ={\left[E_{11},E_{22},2E_{12},E_{33},2E_{23},2E_{13}\right]}^{T}$ and linearized with respect to $\zeta_{3}$ as
(5)$$\epsilon \approx \left[\begin{array}{c} e\left(\zeta_{1},\zeta_{2}\right)+\zeta_{3}\;\chi \left(\zeta_{1},\zeta_{2}\right)\\ E_{33}\left(\zeta_{0}\right)\\ \gamma (\zeta_{1},\zeta_{2}) \end{array}\right]$$
with $\zeta_{0}=\left(\zeta_{1},\zeta_{2},0\right)$ and the membrane strain vector e, the curvature vector χ, and the transverse shear strains vector γ given by
$$e\left(\zeta_{1},\zeta_{2}\right)=\left[\begin{array}{c} E_{11}\left(\zeta_{0}\right)\\ E_{22}\left(\zeta_{0}\right)\\ 2E_{12}\left(\zeta_{0}\right) \end{array}\right],\;\chi \left(\zeta_{1},\zeta_{2}\right)=\left[\begin{array}{c} E_{11}{,}_{3}\left(\zeta_{0}\right)\\ E_{22}{,}_{3}\left(\zeta_{0}\right)\\ 2E_{12}{,}_{3}\left(\zeta_{0}\right) \end{array}\right],\;\gamma \left(\zeta_{1},\zeta_{2}\right)=\left[\begin{array}{c} 2E_{23}\left(\zeta_{0}\right)\\ 2E_{13}\left(\zeta_{0}\right) \end{array}\right].$$

We refer to [13,25] for more details and the explicit strain–displacement relationships. It is important to remember here that, as a consequence of the use of the Green–Lagrange strain measure, quantities $\left\{e\left(\zeta_{1},\zeta_{2}\right),\;\chi \left(\zeta_{1},\zeta_{2}\right),E_{33}\left(\zeta_{0}\right),\gamma \left(\zeta_{1},\zeta_{2}\right)\right\}$ have a quadratic dependence on displacements u.

### 2.2. Constitutive Matrix for the Lamina

Deflections and buckling behavior of multi-layered composites can be modeled efficiently using a homogenized material model based on the hypothesis of the classical lamination theory.

The constitutive matrix of each lamina is usually known with respect to a local Cartesian reference system, defined by the orthogonal triad $\left\{{\bar{\bar{e}}}_{1},{\bar{\bar{e}}}_{2},{\bar{\bar{e}}}_{3}\right\}$, with ${\bar{\bar{e}}}_{1}$ the fibre direction. Then, it is necessary to express it with respect to the Cartesian reference system of the homogenised material, defined by the triad $\left\{e_{1},e_{2},e_{3}\right\}$. Let assume that ${\bar{\bar{e}}}_{3}\equiv e_{3}$ and denote with ϑ the angle around $e_{3}$ between ${\bar{\bar{e}}}_{1}$ and $e_{1}$. To simplify the notation, we omit to report explicitly the dependence of all quantities from the *i*th lamina. Moreover, it is worth noting that ϑ can vary over the mid-surface of the ply in VAT composites.

From now on, symbol $\bar{\bar{\;\;}}$ will denote quantities expressed in components with respect to $\left\{{\bar{\bar{e}}}_{1},{\bar{\bar{e}}}_{2},{\bar{\bar{e}}}_{3}\right\}$. In particular, the Voigt strain vector is $\bar{\bar{\epsilon }}=\left\{{\bar{\bar{e}}}_{p},{\bar{\bar{E}}}_{33},\bar{\bar{\gamma }}\right\}$.

Strain vector $\bar{\bar{\epsilon }}$ is linked to that in the global reference system ϵ by
(6)$$\bar{\bar{\epsilon }}=R\left[\vartheta \right]\epsilon$$
where
(7)$$R\left[\vartheta \right]=\left[\begin{array}{ccc} R_{p} & 0 & 0\\ 0 & 1 & 0\\ 0 & 0 & R_{\gamma } \end{array}\right],$$

0 denotes zero matrices of suitable dimensions and
$$R_{p}=\left[\begin{array}{ccc} \cos{\left(\vartheta \right)}^{2} & \sin{\left(\vartheta \right)}^{2} & -\sin(2\vartheta )\\ \sin{\left(\vartheta \right)}^{2} & \cos{\left(\vartheta \right)}^{2} & \sin(2\vartheta )\\ \sin(2\vartheta )/2 & -\sin(2\vartheta )/2 & \cos{\left(\vartheta \right)}^{2}-\sin{\left(\vartheta \right)}^{2} \end{array}\right]\;\;R_{\gamma }=\left[\begin{array}{cc} \cos(\vartheta ) & \sin(\vartheta )\\ -\sin(\vartheta ) & \cos(\vartheta ) \end{array}\right].$$

The lamina elastic law linking the Green–Lagrange strain to the second Piola–Kirchoff stress is then
(8)S=Cϵ
where by standard transformation
(9)$$C={R[\vartheta ]}^{T}\bar{\bar{C}}R\left[\vartheta \right]=\left[\begin{array}{cccc} C_{p} & 0 & 0 & \\ 0 & C_{33} & 0 & \\ 0 & 0 & C_{t} \end{array}\right].$$

The block matrix $\bar{\bar{C}}$ of the orthotropic constitutive law of the lamina with respect to the lamina reference system is
(10)$$\bar{\bar{C}}=\left[\begin{array}{ccc} {\bar{\bar{C}}}_{p} & 0 & 0\\ 0 & {\bar{\bar{C}}}_{33} & 0\\ 0 & 0 & {\bar{\bar{C}}}_{t} \end{array}\right].$$
where ${\bar{\bar{C}}}_{p}$ is obtained assuming a plane stress condition with a decoupling of membrane and thickness strains in order to eliminate thickness locking [12]. The coefficient ${\bar{\bar{C}}}_{33}$, linking thickness stress and strain, is maintained in order to avoid zero energy modes (thickness stretch).

### 2.3. The Strain Energy for the Shell Model in Generalised Quantities

Letting Ω be the mid-surface of the shell, the stored strain energy of our shell model can be conveniently rewritten, in compact notation, as
(11)$$\Phi \left(u\right)\equiv \frac{1}{2}\int_{\Omega }\int_{-\frac{h}{2}}^{\frac{h}{2}}\epsilon^{T}C\left(\zeta \right)\epsilon \;d\zeta_{3}\;d\Omega =\frac{1}{2}\int_{\Omega }\varepsilon {\left(\zeta_{1},\zeta_{2},u\right)}^{T}𝓒\left(\zeta_{1},\zeta_{2}\right)\varepsilon \left(\zeta_{1},\zeta_{2},u\right)d\Omega$$
with
(12)$$𝓒\left(\zeta_{1},\zeta_{2}\right)=\left[\begin{array}{cccc} {𝓒}_{ee} & 0 & 0 & {𝓒}_{e\chi }\\ 0 & {𝓒}_{33} & 0 & 0\\ 0 & 0 & {𝓒}_{t} & 0\\ {𝓒}_{e\chi }^{T} & 0 & 0 & {𝓒}_{\chi \chi } \end{array}\right]\;\varepsilon \left(\zeta_{1},\zeta_{2},u\right)=\left[\begin{array}{c} e\\ E_{33}\\ \gamma \\ \chi \end{array}\right]$$
where
$${𝓒}_{33}=\int_{-\frac{h}{2}}^{\frac{h}{2}}C_{33}d\zeta_{3}\;{𝓒}_{t}=\int_{-\frac{h}{2}}^{\frac{h}{2}}C_{t}d\zeta_{3}$$
$${𝓒}_{ee}=\int_{-\frac{h}{2}}^{\frac{h}{2}}C_{p}d\zeta \;{𝓒}_{e\chi }=\int_{-\frac{h}{2}}^{\frac{h}{2}}\zeta_{3}C_{p}d\zeta \;{𝓒}_{\chi \chi }=\int_{-\frac{h}{2}}^{\frac{h}{2}}\zeta_{3}^{2}C_{p}d\zeta$$

The transverse shear stiffness ${𝓒}_{t}$ can be evaluated more accurately by means of shear correction factors as, for example, reported in the Abaqus/Standard [37] manual. Finally note that, as recently proposed in [38], thermal effects due to general temperature distributions can be easily accounted for in the model. Generally, a membrane-flexural coupling is possible. Higher order lamination theories or layer-wise interpolations [39,40] are also available for obtaining more accurate inter-laminar stresses. A recent paper [41] proposes the inter-laminar stress recovery starting from the homogenized response.

### 2.4. Discretization Methods

The continuum solid-shell model can be discretized using a displacement based formulation or a mixed one. In this section, a displacement based finite element [12] and isogeometric [14] formulation are reviewed, while we refer to [12,25] for a mixed one.

#### 2.4.1. Geometry and Displacement Interpolation

The shell is discretized in quadrilateral elements through a mesh generation. According to the isoparametric concept, the same interpolation is used for the geometry and displacements. The geometry is described over the element as
(13)$$x\left(\zeta \right)=N_{u}\left(\zeta \right)x_{e}$$
where $x_{e}=\left[x_{0e},x_{ne}\right]$ collects the element discrete variables of the geometry corresponding to $x_{0}$ and $x_{n}$, respectively. The matrix $N_{u}\left(\zeta \right)$ collects the interpolation functions
(14)$$N_{u}\left(\zeta \right):=\left[\begin{array}{c} N\left(\zeta_{1},\zeta_{2}\right),\frac{2\zeta_{3}}{h}N\left(\zeta_{1},\zeta_{2}\right) \end{array}\right]$$
where $N(\zeta_{1},\zeta_{2})$ are bi-dimensional functions of the mid-surface coordinates only as proposed in [13,36].

The displacement field is interpolated using the same shape functions
(15)$$u\left(\zeta \right)=N_{u}\left(\zeta \right)d_{e}\;$$
where $d_{e}=\left[d_{0e},d_{ne}\right]$ collects the element discrete degrees of freedom (DOFs) for the displacement fields $u_{0}$ and $u_{n}$.

The Green–Lagrange strains in Equation (Equation 5), upon considering Equations (Equation 13) and (Equation 15), become
(16)$$\varepsilon \left(\zeta_{1},\zeta_{2},d_{e}\right)=\left(L\left(\zeta_{1},\zeta_{2}\right)+\frac{1}{2}Q\left(\zeta_{1},\zeta_{2},d_{e}\right)\right)d_{e},$$
where $L\left(\zeta_{1},\zeta_{2}\right):=Q\left(\zeta_{1},\zeta_{2},x_{e}\right)$ and $Q(\zeta_{1},\zeta_{2},d_{e})$ has a linear dependence from $d_{e}$, and its expression can be found in [13].

The stored energy of the shell can be evaluated using a numerical integration as
(17)$$\Phi \left(d\right)=\sum_{e}\Phi_{e},\;\Phi_{e}=\frac{1}{2}\sum_{g}\left(\varepsilon_{g}{\left(d_{e}\right)}^{⊤}C_{g}\varepsilon_{g}\left(d_{e}\right)\right)w_{g}$$
where d is the global counterpart of $d_{e}$, *e* denotes the element, *g* indicates the integration point, and $w_{g}$ is the corresponding weight.

#### 2.4.2. Finite Element Formulation

If bilinear shape functions $N(\zeta_{1},\zeta_{2})$ are employed for the middle surface approximation, we have a hexahedron solid-shell linear element. Low order elements are however affected by shear and trapezoidal locking. In order to eliminate these undesired inaccuracies, it is possible to redefine the transverse shear strain components ${\bar{E}}_{\eta \zeta },{\bar{E}}_{\xi \zeta }$ and the transverse normal strain component ${\bar{E}}_{\zeta \zeta }$ by the *Assumed Natural Strain (ANS)* technique with number and location of the sampling points as reported in [12,42]. The in-plane bending response of the element is improved by replacing the in-plane shear strain ${\bar{E}}_{\xi \eta }$ with its value at ξ=η=0 that is a *Selective Reduced Integration (SRI)* retaining the correct matrix rank.

#### 2.4.3. Isogeometric Formulation

In the solid-shell isogeometric version, NURBS of arbitrary order and continuity are employed as middle surface shape functions $N(\zeta_{1},\zeta_{2})$. Locking occurs for low order interpolations. Shear and membrane lockings are typical in small deformation problems. Furthermore, an additional locking occurs in large deformations, even for flat plates, due to the nonlinear strain measure in Equation (Equation 16). The high continuity of the NURBS functions allows the use of patch-wise numerical integrations [43,44], based on a lower number of integration points compared to Gauss rules. Moreover, well-tuned patch-wise reduced scheme can avoid locking. We refer to [13,36,45] for more details on this topic. The displacement-based IGA model represents a reliable choice from the point of view of the discrete approximation and the efficiency of the integration compared to high order FEs.

The NURBS high continuity also allows the use of a Kirchhoff–Love model for thin shells as presented in [45,46], which has the advantage of describing the kinematic using only the mid-surface displacement, thus halving the number of unknowns.

## 3. Objective Function and Design Variables

### 3.1. The Objective Function

The optimization process is aimed at maximizing the collapse load of the composite shells. In buckling problems, the collapse load can be defined as the lower bound between the critical limit load $\lambda_{lim}$ and the load associated with a deformation limit $\lambda_{def}$. With α being the vector collecting the generic design optimization parameters, the objective function can thus be written as
(18)$$P\left(\alpha \right)=\lambda_{c}=\min\left(\lambda_{lim},\lambda_{def}\right).$$

The evaluations $\lambda_{lim}$ and $\lambda_{def}$, and thus the objective function computation, require the construction of the equilibrium path of the structure for assigned design variables α.

Another possibility is to change the optimization problem in minimizing the displacements occurring at an assigned load level. This is particularly suitable for structure with an a priori known stable behavior [35].

### 3.2. Constraints

Some constraints should be considered in the optimization process (see [27]). Firstly, manufacturing constraints are needed to guarantee the applicability for actual engineering products. For example, the manufacturing of VAT composites requires a limit on the minimum steering radius of the fibers. Moreover, the material behavior of the composite can be assumed to be elastic up to failure. In this case, delamination and damage can be prevented by adding constraints on maximum values of the strains or stresses.

### 3.3. Design Variables

The design variables collected in vector α define how the material properties, and then the constitutive matrix in Equation (Equation 12), depend on the material properties at this point, that is, we now have $𝓒(\alpha ,\zeta_{1},\zeta_{2})$.

#### 3.3.1. Layer-Wise Parameters

The most simple approach consists of optimising the lamination using, as optimisation variables, the angles $\vartheta_{i}$ of the stacking sequence, i.e., $\alpha =\{\alpha_{1},\cdots \alpha_{n}\}$ with each $\alpha_{i}\equiv \vartheta_{i}$ constant over the patch. This means that for single shells we have a number of optimization variables at most equal to the number of layers.

For VAT composites, the only change is that the lamina orientation is controlled by more parameters, two according to the description proposed in [47]:$$\vartheta_{i}\left(\zeta_{1},\zeta_{2}\right)=f\left(\zeta_{1},\zeta_{2},\alpha_{k-1},\alpha_{k}\right)\;\mathrm{with}\;k=2i$$

We refer to [27,47] for more details.

#### 3.3.2. Lamination Parameters

The previously discussed layer approach requires a large number of optimization parameters when a large number of layers is used. For this reason, other approaches have been proposed in literature in order to parametrise the homogenised constitutive matrix, without an explicit use of the stacking sequence. To this aim, the main approaches are polar decompositions [48] and lamination parameters [49].

According to the last one, the constitutive matrix can be described as a linear function of 12 lamination parameters [35,50]. Assuming to have the same material through the shell thickness, the matrices ${𝓒}_{ee}$, ${𝓒}_{e\chi }$, ${𝓒}_{\chi \chi }$ and ${𝓒}_{t}$ of the stiffness matrix in Equation (Equation 12), are parametrized in terms of lamination parameters using the material invariants $\Gamma_{k}$:(19)$$\begin{array}{cc} {𝓒}_{ee} & =\Gamma_{0}+\Gamma_{1}\xi_{1}^{A}+\Gamma_{2}\xi_{2}^{A}+\Gamma_{3}\xi_{3}^{A}+\Gamma_{4}\xi_{4}^{A},\;{𝓒}_{e\chi }=\frac{1}{2}\left(\Gamma_{1}\xi_{1}^{B}+\Gamma_{2}\xi_{2}^{B}+\Gamma_{3}\xi_{3}^{B}+\Gamma_{4}\xi_{4}^{B}\right),\\ {𝓒}_{\chi \chi } & =\frac{1}{3}\left(\Gamma_{0}+\Gamma_{1}\xi_{1}^{D}+\Gamma_{2}\xi_{2}^{D}+\Gamma_{3}\xi_{3}^{D}+\Gamma_{4}\xi_{4}^{D}\right),\;{𝓒}_{t}=\Gamma_{0}^{s}+\Gamma_{1}\xi_{1}^{A}+\Gamma_{2}^{s}\xi_{2}^{A}. \end{array}$$

Lamination parameters are defined in [−1,1] as
(20)$$\xi_{i}^{A}=\int_{-1}^{1}f_{i}d\zeta_{3},\;\xi_{i}^{B}=2\int_{-1}^{1}\zeta_{3}f_{i}d\zeta_{3},\;\xi_{i}^{D}=3\int_{-1}^{1}\zeta_{3}^{2}f_{i}d\zeta_{3},\;i=1,\cdots ,4$$
where $f_{i}$ is the component of the vector f=[cos(2ϑ),sin(2ϑ),cos(4ϑ),sin(4ϑ)] and ϑ is the angle around $e_{3}$ of the VAT at a given point. Matrices $\Gamma_{i}$ and $\Gamma_{i}^{s}$ can be evaluated as
(21)$$\begin{array}{c} \Gamma_{0}=\left[\begin{array}{ccc} U_{1} & U_{4} & 0\\ U_{4} & U_{1} & 0\\ 0 & 0 & U_{5} \end{array}\right],\;\Gamma_{1}=\left[\begin{array}{ccc} U_{2} & 0 & 0\\ 0 & -U_{2} & 0\\ 0 & 0 & 0 \end{array}\right],\;\Gamma_{2}=\left[\begin{array}{ccc} 0 & 0 & U_{2}/2\\ 0 & 0 & U_{2}/2\\ U_{2}/2 & U_{2}/2 & 0 \end{array}\right],\;\\ \Gamma_{3}=\left[\begin{array}{ccc} U_{3} & -U_{3} & 0\\ -U_{3} & U_{3} & 0\\ 0 & 0 & -U_{3} \end{array}\right],\;\Gamma_{4}=\left[\begin{array}{ccc} 0 & 0 & U_{3}\\ 0 & 0 & -U_{3}\\ U_{3} & -U_{3} & 0 \end{array}\right], \end{array}$$
(22)$$\Gamma_{0}^{s}=\left[\begin{array}{cc} U_{5} & 0\\ 0 & U_{5} \end{array}\right],\;\Gamma_{1}^{s}=\left[\begin{array}{cc} U_{6} & 0\\ 0 & -U_{6} \end{array}\right],\;\Gamma_{2}^{s}=\left[\begin{array}{cc} 0 & -U_{6}\\ -U_{6} & 0 \end{array}\right],$$
with $U_{k}$ reported in [50].

The lamination parameters, controlling the elastic matrix, are interpolated over the shell domain as
(23)$$\xi =N_{\xi }\left(\zeta_{1},\zeta_{2}\right)\alpha$$
where α collects the lamination parameters $\xi_{i}^{j},\;i=1\ldots 4\;,j=A,B,D$ at the control points of a further grid used for the material description.

Following [51], the reconstruction of the stacking sequences over the surface in terms of the lamination parameters, once they are evaluated by the optimal design process, is obtained in a second stage by minimising a least-square distance between the target distribution (Equation 23) and the lamination parameters related to the unknown fibre angle distribution. To this aim, the vector ϑ collecting the orientations of all layers can be interpolated as
(24)$$\vartheta \left(\zeta_{1},\zeta_{2}\right)=N_{\vartheta }\left(\zeta_{1},\zeta_{2}\right)\vartheta_{d}$$
where $N_{\vartheta }\left[\zeta_{1},\zeta_{2}\right]$ are bi-dimensional shape functions, while $\vartheta_{d}$ are the corresponding discrete variables. The least-square problem can be written as
(25)$$\begin{array}{cccc} \; & \underset{\vartheta_{d}}{\mathrm{minimise}}\; & \; & E\left(\vartheta_{d}\right)=\sum_{i=1}^{n_{p}}\sum_{j=1}^{l_{p}}\frac{{\left(\xi_{j}\left(\vartheta_{d},\zeta_{1}^{i},\zeta_{2}^{i}\right)-\xi_{j}\left(\alpha ,\zeta_{1}^{i},\zeta_{2}^{i}\right)\right)}^{2}}{n_{p}}\\ \; & \mathrm{subject}\;\mathrm{to}\; & \; & \left|\right|\vartheta_{d}{\left|\right|}_{\infty }\leq \pi /2\\ \; & \; & & 𝓒(\vartheta_{d})\leq 0 \end{array}$$
where $𝓒(\vartheta_{d})\leq 0$ is a set of manufacturing constraints that the fibre tow must satisfy, $i=1\ldots n_{p}$ are a fine set of sample points over the shell surface and $l_{p}$ is the number of lamination parameters at each point. The solution of this non-convex and nonlinear problem can be obtained using a multi-start GCMMA. A complex-step method [52] is advised for the gradient evaluations.

## 4. Equilibrium Path Evaluation

The evaluation $\lambda_{lim}$ and $\lambda_{def}$, and thus the objective function computation, requires the construction of the equilibrium path of the structure for assigned design variables α. A common approach for path following the equilibrium curve is the Riks arc-length method [13,16,20,53]. In this case, the nonlinear equations in the kinematic unknowns are solved step-by-step using the Newton–Raphson method. However, this kind of analysis bears a significant computational cost due to the large size of the matrices associated with a high number of DOFs. Furthermore, a reliable evaluation of the equilibrium path should take into account the sensitivity of the structure to imperfections in order to detect the worst imperfection scenario [21]. For this reason, an alternative approach called Koiter’s method [25] was proposed, which assembles a reduced order model based on Koiter’s theory of elastic stability for the assigned material configuration. The corresponding reduced nonlinear equations, usually in a lower number of unknowns, are then solved to obtain a good estimate of the equilibrium path at a low computational cost. The most interesting feature of the method is the possibility of including imperfections a posteriori in the reduced system [28] of the perfect structure, thus enabling inexpensive sensitivity analyses.

### 4.1. Path-Following Analysis

The system of discrete equilibrium equations is then obtained through enforcement of the stationarity of the total potential energy according to
(26)$$r\left(\lambda ;u\right)=\frac{\partial \Phi }{\partial u}-\lambda f=s\left(u\right)-\lambda f=0,$$
where r is the residual vector, s(u) is the vector of generalized stress resultants (i.e., restoring forces), f is the load vector per unit multiplier, u collects the discrete variables and λ is the load multiplier. Note that u collects the global displacements d in a displacement formulation, while it can contain other variables like stresses and strains in mixed formulations. The solutions of Equation (Equation 26) define the equilibrium paths of the structure in the u−λ space.

The Riks approach [16] is the most popular strategy for solving Equation (Equation 26) by adding a constraint of the shape g(λ;u)−s=0, which defines a surface in $R^{N+1}$. Assigning successive values to the control parameter $s=s_{\left(k\right)}$, the solution of the nonlinear system
(27)$$R\left(s\right)\equiv \left[\begin{array}{c} r(\lambda ;u)\\ g(\lambda ;u)-s \end{array}\right]=0$$
defines a sequence of points (steps) $z_{\left(k\right)}\equiv \left\{u_{\left(k\right)},\;\lambda_{\left(k\right)}\right\}$ belonging to the equilibrium path. Starting from a known equilibrium point $z^{0}\equiv z_{\left(k\right)}$, the new one $z_{\left(k+1\right)}$ is evaluated correcting a first *extrapolation*
$z^{1}=\left\{u^{1},\;\lambda^{1}\right\}$ by a sequence of estimates $z^{j}$ (loops) by a Newton iteration
(28)$$\left\{\begin{array}{cc} \; & \bar{J}\dot{z}=-R^{j}\\ \; & z^{j+1}=z^{j}+\dot{z} \end{array}\right.$$
where $R^{j}\equiv R\left(z^{j}\right)$ and $\bar{J}$ is the Jacobian of the nonlinear system (Equation 27) at $z^{j}$ or a suitable estimate. The method is able to provide the equilibrium path for assigned data even in case of limit points in load or displacements. Its main drawback is the high computational cost, due to the large size of the system of equations. For this reason, the method is inadequate to assess the imperfection sensitivity. Generalized path-following methods are a promising alternative to the standard Riks method, where, for example, the critical point can be evaluated by changing the initial data directly, without reevaluating the whole load–displacement curve [20]. However, in the following, a completely different approach, called the Koiter method, is illustrated. It is based on a reducer order model (ROM) and is far more efficient for imperfection sensitivity analyses compared to path-following methods, and then more suitable for design purposes. It is worth citing the possibility to couple the Koiter method with the path-following approach, obtaining the so-called Koiter–Newton approach where the ROM is used as an accurate predictor [29,30].

### 4.2. The Mixed Integration Point Strategy

In geometrically nonlinear analysis, a mixed format of the nonlinear equations in stress and displacement variables provides superior performances in the solution methods. In the standard path-following method, the mixed iterative process assures a greater robustness also for large steps and with a reduced number of iterations, and then a reduced computational time [54]. In Koiter analysis, a mixed format is an indispensable prerequisite to obtain an accurate ROM (see [25,54]). The improved efficiency in path-following methods and accuracy in Koiter’s method is much more evident when the slenderness of the structure gets higher and, concerning the last one, when the pre-critical path presents some nonlinearities. The mixed integration point (MIP) strategy proposed in [17] can be successfully used to exploit the advantages of a mixed format in Riks and Koiter analysis [13,36] without the need of a stress interpolation.

The main idea of the MIP method is to relax the constitutive equations at each integration point by rewriting the strain energy in a pseudo Hellinger–Reissner form as
(29)$$\Phi_{e}\left(u_{e}\right)\equiv \sum_{g=1}^{n}\left(\sigma_{g}^{T}\varepsilon_{g}\left(d_{e}\right)-\frac{1}{2}\sigma_{g}^{T}C_{g}^{-1}\sigma_{g}\right)w_{g}$$
where the stresses at each integration point $\sigma_{g}$ become independent variables:(30)$$u_{e}=\left[\begin{array}{c} \sigma_{1}\\ \vdots \\ \sigma_{n}\\ d_{e} \end{array}\right]$$

The stationary condition with respect to $\sigma_{g}$ gives the constitutive law at *g*:(31)$$s_{g\sigma }\equiv \varepsilon_{g}\left(d_{e}\right)-C_{g}^{-1}\sigma_{g}$$

When Equation (Equation 31) is exactly solved and substituted in Equation (Equation 29), we obtain, again, the displacement formulation. This means that the discrete approximation of the problem is the same as in the original displacement formulation: the equilibrium path is the same when a path-following scheme is adopted (see [17]).

In this way, MIP formulations extend the results already obtained for mixed (stress–displacements) discrete approximations, avoiding the use of a stress interpolation. This is particularly convenient in IGA, where an effective mixed formulation is not an easy task. Moreover, the MIP method was recently used to solve, by means of collocations, the strong form of the problem equations [55,56] or applied to more involved constitutive laws [57].

### 4.3. Koiter Method

The Koiter approach, described in detail in [25,28,36], is here briefly recalled its main algorithmic steps based on the MIP formulation of the solid-shell model. As shown in [54], this is necessary to improve Koiter’s method accuracy because of the direct prediction of the stress and efficiency, due to the vanishing of the fourth order strain energy variations.

By collecting in vector u the global discrete displacements d and stresses $\sigma_{g}$ at each integration point *g*, Koiter’s method is based on the following reduced order model:(32)$$u\left(\lambda ;\psi_{i}\right)=\lambda \hat{u}+\sum_{i=1}^{m}\psi_{i}{\dot{v}}_{i}+\frac{1}{2}\sum_{i,j=1}^{m}\psi_{i}\psi_{j}w_{ij}+\frac{1}{2}\lambda^{2}\hat{\hat{w}}$$
where $\psi_{i}$ are the scalar modal amplitudes, $\hat{u}$ is the linear elastic solution (path tangent to the stress-free configuration), ${\dot{v}}_{i}$ denotes the *i*th of the *n* linearized buckling modes, $w_{ij}$ and $\hat{\hat{w}}$ are quadratic correction modes. The evaluation of these vectors requires the solution of linear systems for $\hat{u}$, $w_{ij}$ and $\hat{\hat{w}}$ and a linearized buckling analysis for ${\dot{v}}_{i}$. Details can be found in [25,28,36].

According to this choice, the equilibrium path is approximated by the following nonlinear reduced system of equilibrium equations in the unknowns $\psi_{i}$λ:(33)$$\begin{array}{cc} r_{k}\left[\lambda ,\psi_{1},\cdots ,\psi_{m}\right] & =\mu_{k}\left(\lambda \right)+\left(\lambda_{k}-\lambda \right)\psi_{k}-\frac{1}{2}\lambda^{2}\sum_{i=1}^{m}\psi_{i}{𝓒}_{ik}+\frac{1}{2}\sum_{i,j=1}^{m}\psi_{i}\psi_{j}A_{ijk}\\ \; & +\frac{1}{6}\sum_{i,j,h=1}^{m}\psi_{i}\psi_{j}\psi_{h}B_{ijhk}=0,\;k=1\ldots m \end{array}$$

The scalar coefficients $A_{ijk}$, $C_{ik}$, $B_{ijhk}$ and $\mu_{k}\left[\lambda \right]$ are computed as the sum of elemental contributions of the stored energy variations. Their explicit expressions can be found in [25,36].

A notable feature of the method is a computationally efficient imperfection sensitivity analysis. In fact, the imperfect structure can be studied by perturbing a posteriori the same reduced system of the perfect structure by adding to it the imperfection coefficients ${\tilde{\mu }}_{k}$:(34)$$r_{k}+{\tilde{\mu }}_{k}=0,\;k=1\ldots m.$$

This means the analysis of a new geometrical imperfection simply requires only to update ${\tilde{\mu }}_{k}$ and solve again the small system in Equation (Equation 34). Thousands of imperfections can be analyzed in a few seconds regardless of the global number of DOFs used for the full structural model.

Two strategies were proposed for the evaluation of ${\tilde{\mu }}_{k}$ [28]. A first solution is very quick but with a validity range restricted to small imperfection amplitudes and almost linear pre-buckling path. A second imperfection modeling is more accurate for a wider range of structural problems and just a little more expensive than the first one [28].

#### 4.3.1. The Worst-Case Geometrical Imperfection

The geometrical imperfection $\tilde{d}$ can be assumed to be a linear combination of assigned shapes ${\dot{d}}_{i}$ with combination factors $\tilde{\psi_{i}}$,
(35)$$\tilde{d}=\sum_{i=1}^{m}\tilde{\psi_{i}}{\dot{d}}_{i},\;||\tilde{d}\left|\right|\leq {\tilde{d}}_{max},$$
scaled in order to have an assigned maximum amplitude ${\tilde{d}}_{max}$ chosen, for example, from experimental measurements as in [58]. Shapes ${\dot{d}}_{i}$ can be chosen, for example, as the displacement part of the first linearized buckling modes.

The worst-case imperfection can be defined as that leading to the worst value of $P(\alpha_{g},{\tilde{\psi }}_{1},\ldots ,{\tilde{\psi }}_{m})$:(36)$$\begin{array}{cccc} \; & \underset{{\tilde{\psi }}_{1},\ldots ,{\tilde{\psi }}_{m}}{\mathrm{maximise}}\; & \; & P[\alpha_{g},{\tilde{\psi }}_{1},\ldots ,{\tilde{\psi }}_{m}]\\ \; & \mathrm{subject}\;\mathrm{to}\; & \; & {\left|\tilde{d}\left[{\tilde{\psi }}_{1},\ldots ,{\tilde{\psi }}_{m}\right]\right|}_{\infty }={\tilde{d}}_{max} \end{array}$$

The solution of the previous problem can be obtained by the stochastic algorithm proposed by [26,27], which exploits the reduced order modeling of Koiter’s method.

#### 4.3.2. Advantages of a TL Solid-Shell Formulation

Discrete models directly derived from the 3D continuum using the Green strain measure have a low order dependence on the strain energy from the discrete parameters and in particular have a 3rd order dependence for the MIP formulation. On the contrary, geometrically exact shell and beam models or those based on corotational approaches make use of the rotation tensor. This implies that the stored energy is infinitely differentiable with respect to the discrete parameters and leads to very complex expressions for the energy variations with a high computational burden for their evaluation. On the contrary, for solid-shell finite elements, the strain energy in MIP form has the lowest polynomial dependence on the corresponding discrete parameters, i.e., just one order more than in the linear elastic case, implying the zeroing of the fourth order energy variations required to build the Koiter ROM.

## 5. Postbuckling Optimisation Algorithms

The optimisation problem is based on searching for the material distribution that maximizes the structural performance, i.e., the objective function in Equation (Equation 18).

### 5.1. Monte Carlo Random Search with Zoom Steps

The Monte Carlo random search method proposed in [26] consists of multiple steps and requires only the computation of the objective function. During the first step, a random population of $N_{1}$ layups is generated and, for each of them, the objective function is computed. The $n=n_{1}$ elite (best) solutions, collected in $\alpha_{el}$, represent the starting points of the second step (zoom step) that improves the previous elite. For each value in $\alpha_{el}$, the objective function is evaluated $N_{2}$ times at random points defined as
(37)$$\alpha_{j}=\alpha_{el}\left(j\right)+rnd\left(-R,R\right)$$
where j=1…n denotes the elite value and rnd is a generator of pseudo random integer values between −R and *R*. The radius *R* can decrease during the steps, e.g., $R_{1}$ during the first zoom step and $R_{2}$ for the others.

At the end of a zoom step, *n* best solutions are selected as the new elite population to start the next step. The algorithm stops if a satisfactory convergence is obtained; otherwise, a next zoom step is started. Although very simple, the Monte Carlo search with zoom steps provides good estimates of the optimum, for practical design purposes, with a limited number of objective function evaluations. These kinds of methods become, however, more and more demanding by increasing the number of design variables.

### 5.2. Genetic Algorithm

A genetic algorithm (GA) is a metaheuristic method inspired by the natural selection [32,59]. It is widely used to solve optimization problems when the derivatives of the objective function are difficult or costly to evaluate. Compared to a Monte Carlo random search, it relies on biologically inspired operators such as mutation, crossover and selection. The method starts from a random population and proceeds with an iterative process, called generation, providing new individuals. At each generation, the value of the objective function (called fitness) of every individual is evaluated. The best individuals are stochastically selected and recombined also with random mutations to form a new generation for the next iteration. The termination criterion is a maximum number of generations or a satisfactory fitness level. In addition, the genetic algorithm becomes, however, costly for a large number of design variables.

### 5.3. Globally Convergent Method of Moving Asymptotes

The optimal design problem can be also solved via a gradient based method with convex subsequent approximations of the objective function, i.e., the Global Convergent Method of Moving Asymptotes (GCMMA) [33,60,61,62]. It is particularly suitable for the optimisation of objective functions requiring a high computational cost and depending on many variables.

The gradient evaluation with respect to the design variables is usually done numerically. In particular, the *i*th component is computed as
(38)$$\nabla P_{i}\left(\alpha \right)\approx \frac{P\left(\alpha +hi_{i}\right)-P\left(\alpha \right)}{h}$$
where *h* is a small real parameter chosen defining the discrete incremental ratio, and $i_{i}$ is a basis vector where the unitary *i*th component is the only non-zero element. Generally, the gradient evaluation is extremely time-consuming, but the relative efficiency of Koiter’s method reduces it to an acceptable time. Moreover, GCMMA generally converges with a few iterations, so compensating largely the gradient computation. A multi-start GCMMA is advised for complicated objective functions with multiple local optima.

### 5.4. Comparison of the Optimization Algorithms

According the numerical experience matured in [26,27,35] in post-buckling optimisation, the performances of the different algorithms can be compared. In particular, the main advantage of Monte Carlo with zoom steps and genetic algorithm is a likely convergence to the global optimum and no need for gradient computation. The first one usually provides better estimates for a low number of function evaluations, but it is slower to achieve a converged solution compared to the second one. However, both methods get inefficient (large number of function evaluations) when the number of design parameters increases. On the contrary, GCMMA is the most efficient choice for a large number of design variables because the cost for computing the function gradient is largely compensated by the very fast convergence. As a drawback, multiple starts are generally needed to assure convergence to global optima.

## 6. Post-Buckling Optimisation of a Cylindrical Panel under Compression

Figure 2 depicts a simply-supported cylindrical panel subject to uniform axial compression. Boundary conditions are reported in Figure 2 and imposed as usual by zeroing the corresponding DOFs. For the axial displacement, the rigid motion is prevented by maintaining the symmetry.

A postbuckling optimisation of this structure was previously carried out in [26] using straight fibre laminates and a Monte Carlo methodology. In addition, it was used as a benchmark test for the isogeometric formulation proposed in [13]. The panel is particularly suitable to assess the potentiality of the material optimisation, since its baseline Quasi-Isotropic (QI) design, that is, when all lamination parameters are zero, exhibits an unstable post-buckling behaviour with imperfection sensitivity. In the following, we will denote with $n_{1}\times n_{2}$ the number of isogeometric elements in the two directions of the shell domain. The NURBS isogeometric discretization is adopted for both material description and displacement field. The number of elements for the two discretizations is not necessarily the same. In particular, for the structural analysis, the displacement discretization is that giving converged results, while, for the material description, different meshes are considered.

### 6.1. Stage 1: Determination of the Optimum in Terms of Lamination Parameters

The optimisation problem at stage 1 consists of finding the distribution of lamination parameters that maximises the failure load, i.e., the minimum between the first limit load $\lambda_{lim}$ and the load leading to a limit axial displacement of the loaded section $v_{ax}=v_{d}$. In practice, the process searches for the axially-stiffest stable behaviour or the unstable configuration with the highest limit load and is less sensitive to geometrical imperfections. The loads in the objective function (Equation 18) are normalised by the first linearised buckling load of the QI case, named $\lambda_{QI}$.

The shell thickness is t=10 mm. The axial displacement limit is set to $v_{d}=2$ mm. The material is a widely employed E-glass/epoxy fibrecomposite with $E_{11}=30.6$ GPa, $E_{22}=8.7$ GPa, $\nu_{12}=0.29\;$, $\nu_{23}=0.5\;$, $G_{12}=3.24$ GPa, $G_{23}=2.9$ GPa. The discretization of the displacement field is based on quadratic NURBS and a 9×9 control grid that gives a converged solution for different material configurations. Consequently, the number of discrete displacement DOFs is 683. The maximum amplitude of the geometrical imperfection is ${\tilde{d}}_{max}=0.5$ t.

Two different grids based on quadratic NURBS, namely 6×6 and 9×9, are used to parametrize the lamination parameters. To preserve the symmetry, only the control points of a quarter of the structure are taken as independent design parameters. An orthotropic laminate with symmetric and balanced stacking sequence is considered. Under these choices, the number of independent design variables is restricted to 64 and 144, respectively, for the two grids.

The convergence of GCMMA for the two different description of the lamination parameters is illustrated in Figure 3. The optimal solution is achieved by less than 10 iterations. Moreover, a good solution is also given for the coarsest material mesh. The evolution of the equilibrium path during the iterative process is shown in Figure 4 for the 9×9 material grid. Each equilibrium curve corresponds to the lamination parameters obtained at the end of each GCMMA iteration, whose number is reported in the legend. For example, the equilibrium path indicated with 0 corresponds to the starting point of GCMMA, which is the QI configuration. The curve denoted with 1 is relative to the first iteration of GCMMA, and so on. Interestingly, the solution gets significantly better already after the first iteration compared to the initial QI configuration. The third iteration gives the first stable postbuckling response. Then, the computational design process finds configurations with increasing axial stiffness and the GCMMA algorithm converges soon afterwards: the equilibrium curve after the 7th iteration is practically the same as that found at the 20th one.

The results of stage 1 are summarized in Figure 5 in terms of equilibrium curves. In particular, the load–displacement path corresponding to the optimal distribution of lamination parameters (LP) is compared with the baseline QI case. Comparison is also made with the optimal solution for straight fibre laminates with a non-symmetric stacking sequence (SF) obtained in [26], that is, from the inside out, $\left[\mp 54,0_{4}\right]$ with the fibre orientations given with respect to the local reference system in Figure 2 with $e_{3}$ aligned with the surface normal vector. Figure 5 compares the equilibrium path predicted by Koiter’s method with that given by the Riks method using the full model. The end shortening and the out-of-plane displacement at the centre of the panel are monitored. We can observe a satisfactory agreement in all cases. The optimal configuration of lamination parameters is illustrated in Figure 6. Finally, the worst-case imperfection detected during the optimal design process is pictured in Figure 7.

### 6.2. Stage 2: Recovery of the Lamination Angles

The advantage of the two-stage methodology is the possibility of designing structures made of many layers without penalizing the computational cost of the process. Actually, having many layers is an advantage, since it allows a more accurate match with the optimal solution of stage 1. This fact is shown here, where the recovered stacking sequence (SS) is symmetric and balanced with 12 independent layers. The stacking sequence is restricted to be ${\left[\pm \theta_{1},\ldots ,\pm \theta_{12}\right]}_{s}$, with each θ interpolated using a bivariate quadratic NURBS (see Equation (Equation 24)) over a mesh of 9×9 elements. The thickness of each layer is 0.208 mm.

The solution of the problem (Equation 25) is achieved by means of a multi-start GCMMA. The algorithm converges quickly, as shown in Figure 8. The corresponding equilibrium curves are reported in Figure 9, while Table 1 shows the match between the values of objective function and the first linearised buckling loads for the optimal lamination parameters and those corresponding to the recovered fibre orientations. These last ones are pictured in Figure 10.

## 7. Conclusions

This review collected some recent findings in the post-buckling optimisation of thin-walled composite structures. Focus was given to structural modelling, material parametrization, post-buckling analysis with imperfection sensitivity and optimisation algorithms. A computational framework merging the developments in these different aspects led to robust and efficient optimizations of composite materials with spatially varying material proprieties. As a result, numerical analysis can make it possible to fully exploit the capability of advanced manufacturing methods for the realisation of a new generation of structures able to work safely in the post-buckling regime saving materials, costs and weight. 

## Figures and Tables

**Figure 1 materials-14-01665-f001:**
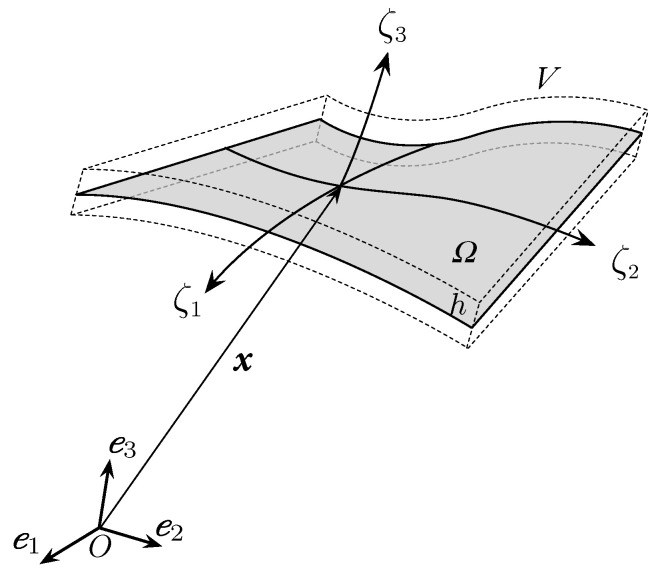
Schematic geometry of the solid-shell continuum.

**Figure 2 materials-14-01665-f002:**
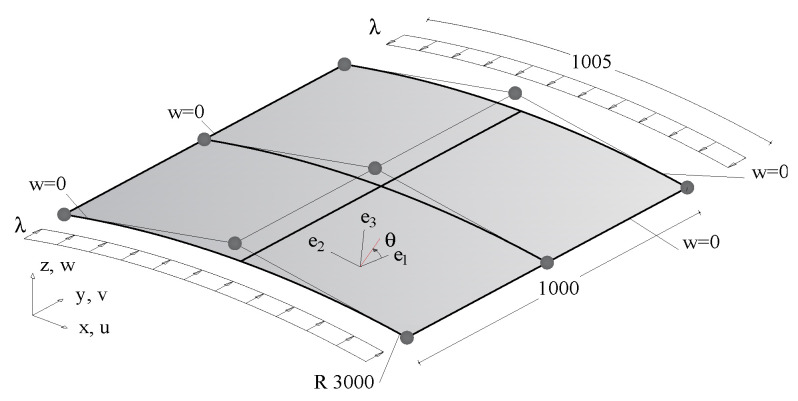
Geometry (lengths in mm), NURBS control grid, boundary conditions and loads.

**Figure 3 materials-14-01665-f003:**
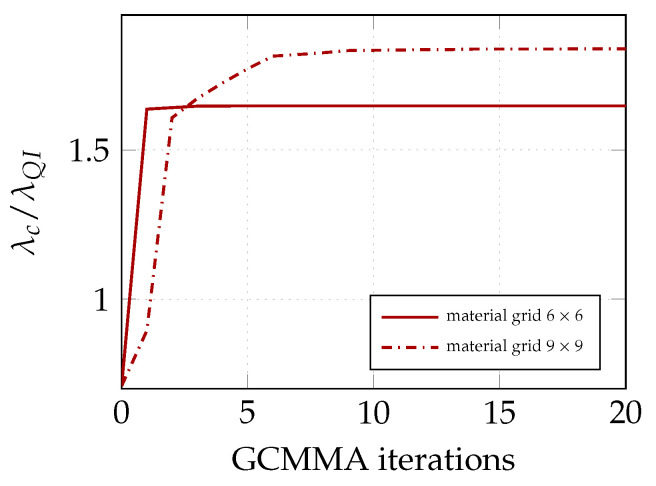
Convergence of the GCMMA optimisation algorithm at stage 1 for two different NURBS parameterizations of the lamination parameters.

**Figure 4 materials-14-01665-f004:**
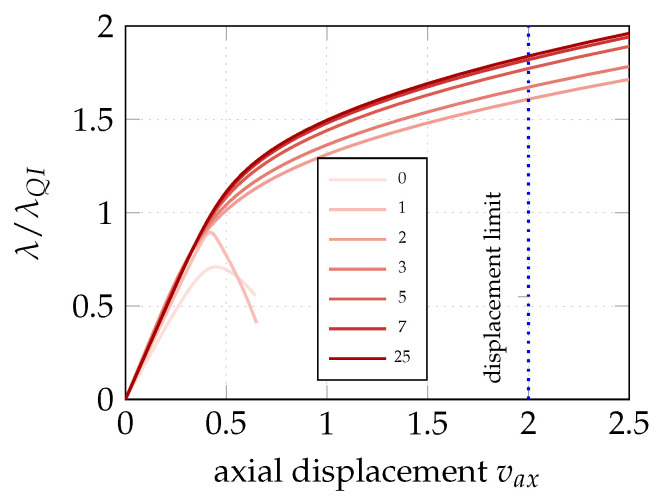
Load–displacement paths after different GCMMA iterations (0,1,…25) at stage 1 with a grid of 9×9 material control points.

**Figure 5 materials-14-01665-f005:**
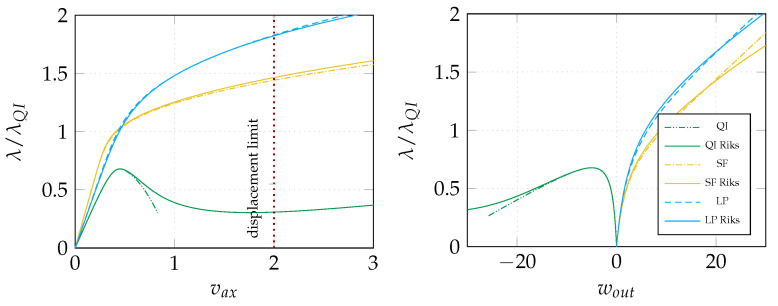
Comparison of the equilibrium paths obtained using Koiter’s method with Riks results. On the left, the axial displacement ($v_{ax}$) is plotted while, on the right, we have the out-of-plane displacement at the panel center ($w_{out}$). Baseline QI case, straight fibre design [26] and optimal lamination parameters are considered.

**Figure 6 materials-14-01665-f006:**
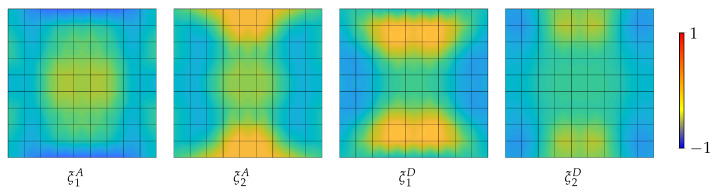
Optimal distribution of lamination parameters over the shell domain for the material grid of 9×9 elements [35].

**Figure 7 materials-14-01665-f007:**
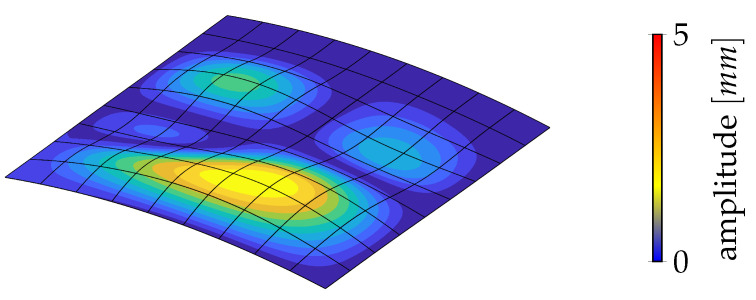
Worst-case geometrical imperfection detected in stage 1 for the optimal distribution of lamination parameters [35].

**Figure 8 materials-14-01665-f008:**
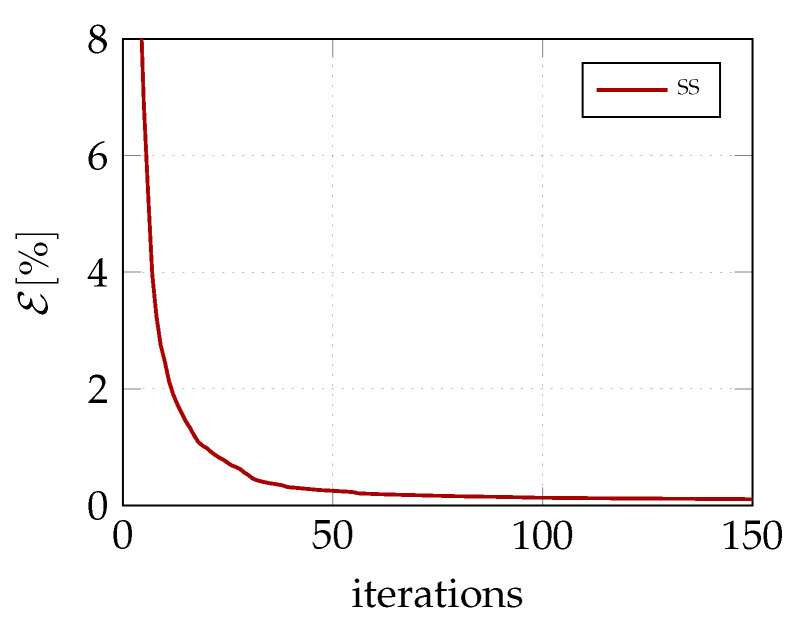
Convergence of the multi-start GCMMA for the optimisation problem of stage 2.

**Figure 9 materials-14-01665-f009:**
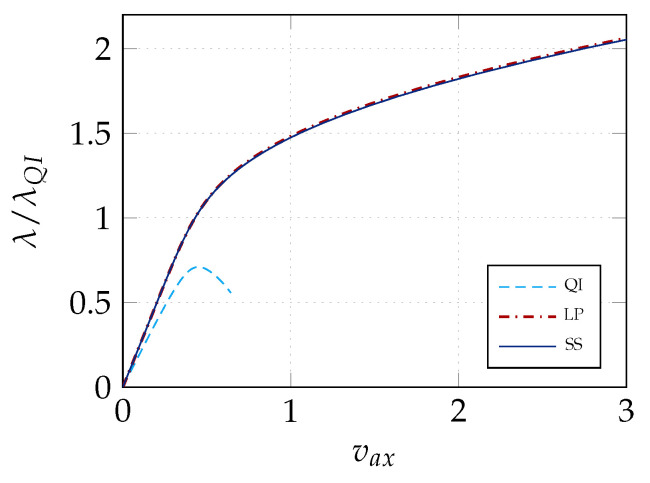
Load-axial displacement $v_{ax}$ paths for the baseline QI case, the optimal distribution of lamination parameters (LP, 9×9 elements) and the fibre orientations retrieved in stage 2 (SS).

**Figure 10 materials-14-01665-f010:**
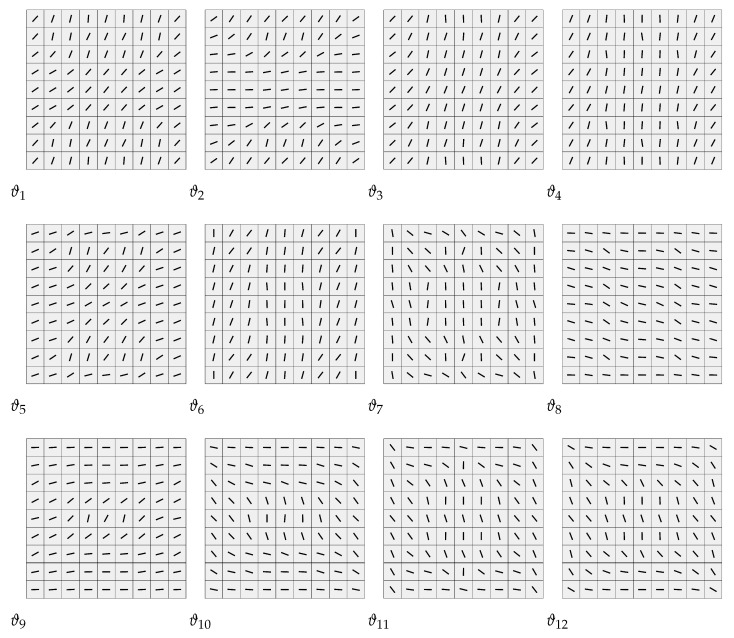
Fibre orientations retrieved by stage 2 for the stacking sequence ${\left[\pm \vartheta_{1},\pm \vartheta_{2},\ldots ,\pm \vartheta_{12}\right]}_{S}$ [35].

**Table 1 materials-14-01665-t001:** Results of the optimised material design LP for a grid of 9×9 control points for the lamination parameters and for the retrieved fibre orientations (SS).

	9×9					
Case	P	$\lambda_{1}$	$\lambda_{2}$	$\lambda_{3}$	$\lambda_{4}$	Match
QI	0.710	1.000	1.255	1.795	1.883	-
LP	1.830	1.177	1.489	1.840	2.100	-
SS	1.821	1.170	1.482	1.840	2.097	99.92%

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
