# Peer review of "Material Design for Optimal Postbuckling Behaviour of Composite Shells"

_materials, 2021, doi:10.3390/ma14071665_

Round 1

Reviewer 1 Report

This review paper shows recent research developments of material design for optimal post-buckling behaviour of composite shells with computational approach. Well written, shown and investigated.

Although the authors show the scalability of post-buckling optimization of composite material, they should point out the applicability for actual engineering use or products with productivity. Activity of buckling design for composite thin shell should be explained.

Also, almost composite material has elastic property without plasticity/ductile stress-strain relation. Material non-linearity or damages should be noted.

It is recommended that authors should refer some experimental research, because 3D printed shells with fiber reinforced plastics can be made in recent.

Author Response

This review paper shows recent research developments of material design for optimal post-buckling behaviour of composite shells with computational approach. Well written, shown and investigated.

 Although the authors show the scalability of post-buckling optimization of composite material, they should point out the applicability for actual engineering use or products with productivity. Activity of buckling design for composite thin shell should be explained.

 Also, almost composite material has elastic property without plasticity/ductile stress-strain relation. Material non-linearity or damages should be noted.

 It is recommended that authors should refer some experimental research, because 3D printed shells with fiber reinforced plastics can be made in recent.

  • We added some comments on already published works regarding actual aerospatial products designed against buckling using the proposed approach and taking into account manufacturing constraints at the end of the introduction.
  • In the optimization design process maximum strains are usually controlled in order to avoid delamination or damage. We added a short subsection 3.2 on this point
  • We added some references on recent experimental researches on 3D printed shells with reinforced plastics at the beginning of the introduction.

Reviewer 2 Report

This work is a review of the computational developments concerning the optimisation of the response of composite thin-walled structures prone to buckling, showing how baseline products with unstable behaviour can be transformed in stable ones operating safely in the post-buckling range.

The work describes the stage of numerical analisis in a laconic manner. Therefore, please respond to the following comments:

  1. The description of the manuscript lacks a description of what material was used and how the mechanical properties presented in Tab. 1.
  2. On page 15, line 359 the authors describe the size of the mesh, missing the element size unit.
  3. What kind of finite element was used during the numerical analysis.
  4. How was the composite numerically modeled, the results of which are presented e.g. in Fig. 7.
  5. How the boundary conditions were numerically defined.
  6. Fig. 3 shows the influence of the finite element mesh on the behavior of the system. In my opinion, the size of the element influences the buckling behavior of the structure. We propose to deal with the 6x6 variant in the future.
  7. Figures 4-5 show the equilibrium paths. How are the initial imperfections of the structure defined (initial deviation of the path from the vertical axis of the graph).

Author Response

This work is a review of the computational developments concerning the optimisation of the response of composite thin-walled structures prone to buckling, showing how baseline products with unstable behaviour can be transformed in stable ones operating safely in the post-buckling range.

The work describes the stage of numerical analisis in a laconic manner. Therefore, please respond to the following comments:

  1. The description of the manuscript lacks a description of what material was used and how the mechanical properties presented in Tab. 1.
  2. On page 15, line 359 the authors describe the size of the mesh, missing the element size unit. What kind of finite element was used during the numerical analysis.
  3. How was the composite numerically modeled, the results of which are presented e.g. in Fig. 7.
  4. How the boundary conditions were numerically defined.
  5. Fig. 3 shows the influence of the finite element mesh on the behavior of the system. In my opinion, the size of the element influences the buckling behavior of the structure. We propose to deal with the 6x6 variant in the future.
  6. Figures 4-5 show the equilibrium paths. How are the initial imperfections of the structure defined (initial deviation of the path from the vertical axis of the graph).
  1. We added the information required on the material in the caption of Tab. 1.
  2. We clarified the notation adopted: 6x6 means a grid with 6 subdivisions per edge (36 elements). This has been specified in section 2, but we highlighted now this information also at the beginning of the numerical results section. In particular no finite element are used but an IGA discetization.
  3. The description of the modeling of the composites has been presented in section 2.
  4. Boundary condition are depicted in Fig. 2 and imposed as usual by zeroing the corresponding DOFs. For the axial displacement, the rigid motion is prevented by maintaining the symmetry. This sentence was added at the beginning of the test.
  5. These grids are not for the structural discretization but only for the description of the lamination parameters over the shell surface. The associated discrete variables describe how the material properties vary over the domain and represent the variable to be optimized. For the structural analysis a different discretization is adopted, i.e. leading to converged results. This is already explained in the text.
  6. Imperfections are geometrical deviations defined as in subsection 4.3.1 and modelled as in Eq. 34 as better explained in Ref.28 which was already cited on this point.

Reviewer 3 Report

The present paper is an interesting review of some recent findings in the post-buckling optimization of thin-walled composite structures. The authors are kindly requested to correct or comment some details of minor importance:

  1. The symbols and E (below line 127) do not correlate with the corresponding symbols of Eq. (6).
  2. A brief description is needed for the symbol Ω of Eq. (11).
  3. A few typographical errors should be corrected: line 195 (We refers …), line 216 (… with a the higher number…), line 268 (… of the forth fourth order…).
  4. Numbering is needed for the first equation of page 13 of 23 (above line 304).
  5. A reference should be made for the genetic algorithm presented in the subsection 5.2.
  6. Since this is a review article, a more detailed comparison among the post-buckling optimization algorithms of section 5 should be made.

Author Response

The present paper is an interesting review of some recent findings in the post-buckling optimization of thin-walled composite structures. The authors are kindly requested to correct or comment some details of minor importance:

  1. The symbols and E (below line 127) do not correlate with the corresponding symbols of Eq. (6).
  2. A brief description is needed for the symbol Ω of Eq. (11).
  3. A few typographical errors should be corrected: line 195 (We refers …), line 216 (… with a the higher number…), line 268 (… of the forth fourth order…).
  4. Numbering is needed for the first equation of page 13 of 23 (above line 304).
  5. A reference should be made for the genetic algorithm presented in the subsection 5.2.
  6. Since this is a review article, a more detailed comparison among the post-buckling optimization algorithms of section 5 should be made.

  • We corrected the symbol. Thank you
  • We agree. The description has been added
  • Thank you, we corrected the typos.
  • We added the number to the equation
  • A Reference has been added
  • We agree. A comparison of the optimization algorithms have been added.